# Comparison of tests done, and tuberculosis cases detected by Xpert® MTB/RIF and Xpert® MTB/RIF-Ultra in Uganda

**Michael Kakinda**[1]*, **Didas Tugumisirize**[2], **Abdunoor Nyombi**[2], **Marvin Mugisha**[2], **Stavia Turyahabwe**[2], **Simon Walusimbi**[3,4], **Joseph K. B. Matovu**[5,6]

**1** Joint Clinical Research Center, Kampala, Uganda, **2** National Tuberculosis and Leprosy Program, Ministry of Health, Kampala, Uganda, **3** Makerere University Lung Institute, Makerere University, Kampala, Uganda, **4** Makerere University College of Health Sciences, Makerere University, Kampala, Uganda, **5** Makerere University School of Public Health, Kampala, Makerere University, Kampala, Uganda, **6** Busitema University Faculty of Health Sciences, Busitema University, Mbale, Uganda

* michaelkakinda@gmail.com

## Abstract

### Background

Uganda introduced Xpert® MTB/RIF assay into its TB diagnostic algorithm in January 2012. In July 2018, this assay was replaced with Xpert® MTB/RIF Ultra assay. We set out to compare the tests done and tuberculosis cases detected by Xpert® MTB/RIF and Xpert® MTB/RIF Ultra assay in Uganda.

### Methods

This was a before and after study, with the tests done and TB cases detected between Jan-June 2019 when using Xpert® MTB/RIF Ultra assay compared to those done between Jan-June 2018 while using Xpert® MTB/RIF assay. This data was analyzed using Stata version 13, it was summarized into measures of central tendency and the comparison between Xpert® MTB/RIF Ultra and Xpert® MTB/RIF was explored using a two-sided T-test which was considered significant if p <0.05.

### Results

One hundred and twelve (112) GeneXpert sites out of a possible 239 were included in the study. 128,476 (*M*: 1147.11, SD: 842.88) tests were performed with Xpert® MTB/RIF Ultra assay, with 9693 drug-susceptible TB (DS-TB) cases detected (*M*: 86.54, SD: 62.12) and 144 (M: 1.28, SD: 3.42) Rifampicin Resistant TB cases (RR-TB). Whilst 107, 890 (*M*: 963.30, SD: 842.88) tests were performed with Xpert® MTB/RIF assay between, 8807 (M: 78.63, SD: 53.29) DS-TB cases were detected, and 147 (M: 1.31, SD: 2.39) RR-TB cases. The Number Need to Test (NNT) to get one TB case was 12 for Xpert® MTB/RIF and 13 for Xpert ®MTB/RIF Ultra. On comparing the two assays in terms of test performance (p = 0.75) and case detection both susceptible TB (p = 0.31) and RR-TB (p = 0.95) were not found statistically significant.

**Data Availability Statement:** All relevant data are within the paper and its Supporting Information files.

**Competing interests:** The authors have declared that no competing interests exist.

## Conclusions

This study found no significant difference in test performance and overall detection of DS-TB and RR-TB when using Xpert® MTB/RIF Ultra and Xpert® MTB/RIF assays. The health systems approach should be used to elucidate all the probable potential of Xpert® MTB/RIF Ultra.

## Background

The World Health Organization (WHO), in 2010, endorsed the Xpert® MTB/RIF assay (Xpert) (Cepheid, Sunnyvale, CA, USA) a nested real-time PCR, which can simultaneously detect DNA of *Mycobacterium tuberculosis (MTB)* and Rifampicin resistance within 2 hours [1]. Whilst it shows excellent sensitivity of 98% in the diagnosis of pulmonary TB (PTB) with smear-positive sputum, the sensitivity of Xpert® MTB/RIF for PTB detection with smear-negative sputum (67%), in HIV positive participants (81%), and children (62%) is considered suboptimal [2–5]. Furthermore, the sensitivity of Xpert® varies across different extrapulmonary specimens, ranging from 50.9% in the pleural fluid to 97.2% in bone or joint fluid [6–8].

To overcome these limitations Xpert® MTB/RIF Ultra assay (Xpert® Ultra) was developed and endorsed for use by WHO in 2017. Xpert® MTB/RIF Ultra assay has several advantages over Xpert® MTB/RIF assay, it has shown an 11–17% increase in sensitivity among smear-negative, culture-positive samples compared to the Xpert® MTB/RIF assay [3, 4]. The Xpert® Ultra assay also has a shorter run time of 77 minutes per positive sample and 65 minutes for a negative sample when compared to the Xpert® MTB/RIF assay's 114 minutes [9]. Therefore, by simply switching from Xpert® MTB/RIF to Xpert® MTB/RIF Ultra assay, programs could increase their already installed GeneXpert® system capacity by up to 50% depending on the positivity rate [10].

Uganda introduced Xpert® MTB/RIF assay into its TB diagnostic algorithm in January 2012. In July 2018, due to the above-mentioned reasons, this assay was replaced with the Xpert® MTB/RIF Ultra assay. We set out to determine, the difference in the number of tests done and TB cases detected by Xpert® MTB/RIF Ultra in comparison with Xpert® MTB /RIF.

## Methods and materials

### Study design and setting

This was a before and after study, where GeneXpert® tests and TB cases detected between January to June 2019 were compared to those done between January to June 2018 when Xpert® MTB/RIF Ultra and Xpert® MTB/RIF were used, respectively. In 2018, Uganda had 239 GeneXpert® sites randomly distributed across the then 122 districts of Uganda.

The country's health system is stratified according to levels of care from top to bottom as follows; National Referral Hospitals (NRHs), Regional Referral Hospitals (RRHs), General district hospitals (GHs), Health Centre IVs (HCIVs), Health Centre IIIs (HCIIIs) and Health Centre IIs (HCIIs). GeneXpert services are however limited to both regional and district General hospitals and some selected HCIVs and HCIIIs based on infrastructural requirements, workload, and accessibility among other factors. The country has a robust sample referral system (HUB system) where samples are picked from peripheral health facilities without GeneXpert machines to functional GeneXpert sites and the results are relayed back the same way [11].

## Study population

One hundred and twelve (112) GeneXpert sites out of a possible 239 were included in the study. All the GeneXpert sites in the country did transition from Xpert MTB/RIF to Xpert MTB/RIF Ultra. GeneXpert sites that had a weekly reporting rate of 100% between January-June 2018 and January to June 2019 were included in this study, the same sites should have reported for both periods, therefore the sites were matched for both periods. They were excluded if either they did not report to the National TB and Leprosy Program (NTLP) or they broke down, hence reporting zeros.

## Study description

Weekly, all GeneXpert sites in the country report GeneXpert surveillance data to the National TB and Leprosy Program. This data is collected through a standardized template that has a number of tests done, number with Drug-Sensitive TB (DS-TB), number with Rifampicin Resistant TB (RR-TB), number of children tested, and errors codes.

The GeneXpert sites that don't report are flagged by the focal person at the National TB Reference Laboratory (NTRL) and later reminded to do so. If they still don't report, then the Regional TB and Leprosy Focal Person (RTLFP) and Regional Implementing Partner (IP) are tasked with ensuring that reporting is done. These reports are then collated and presented to the various stakeholders (Ministry of Health, Development Partners, Donors, and Implementing Partners) as utilization rates (Number of tests done per day) per site. The sites with a good GeneXpert Utilization (>16 tests done per day) are applauded, while the poorly performing sites (usually < 4 tests a day) are tasked to explain the performance and improve. This data is then collated into quarterly and annual reports which make part of the annual performance report for the NTLP.

## Data collection methods and procedure

We used programmatic data generated from the weekly GeneXpert surveillance reports routinely submitted to the National TB programs by GeneXpert sites with support from regional Implementing partners through emails, GX-alert systems, or phone calls. This surveillance data is collected through a standardized template that has a number of tests done, number with Drug-Sensitive TB (DS-TB), number with Rifampicin Resistant TB (RR-TB), and number of children tested, and errors codes. We collated data from the weekly GeneXpert surveillance reports into a spreadsheet (Windows Excel 2016, Microsoft Corp., Redmond, WA). The data was then subjected to the inclusion and exclusion criteria, then checked for inconsistencies, and cleaned in preparation for analysis.

## Study outcomes

The primary outcomes were the tests and TB cases detected by Xpert® MTB/RIF Ultra and Xpert® MTB/RIF assays. While the secondary outcomes were Number Needed Test (NNT) to get one TB case, and a comparison between tests done and the number of TB cases detected (DS-TB and RR-TB) by Xpert® MTB/RIF Ultra and Xpert® MTB/RIF assays respectively.

## Data analysis

We exported data from a spreadsheet (Microsoft Excel version 2013) into Intercooled Stata version 13 (Stata-Corp, College Station, Texas, USA). The outcome data were summarized into measures of central tendency (Standard deviation and mean) and the comparison

between Xpert® MTB/RIF Ultra and Xpert® MTB/RIF was explored using a two-sided T-test which was considered significant if p <0.05.

## Ethical approval

The study used secondary data which was anonymous and widely available. However, approval for the study was obtained from the Operational Research and Ethics Review Committee of the Uganda National Tuberculosis and Leprosy Control Programme.

## Results

### Description of the study sites

One hundred twelve (112) sites were included in the study and were distributed as follows according to the Ministry of Health Uganda Regions, which is based on the Regional Referral Hospital the facilities fall under (Kampala-15, Mubende-13, Lira-5, Gulu-8, Arua-9, Moroto-6, Masaka-6, Mbarara-15, Fort Portal-7, Hoima-6, Jinja-8, Mbale-7, and Soroti-7). During the periods before and after switching from Xpert MTB/RIF to Xpert MTB/RIF Ultra, more tests were done by Mbarara Region but most cases were detected by Kampala Region, while Lira Region did the least number of tests when MTB/RIF was being used and detected the least cases, while on switching to MTB/RIF Ultra it was Mbale Region with both the least number of tests and cases detected (See Table 1).

The health facilities were distributed as follows according to levels of care; 45 General Hospitals (GH), 34 Health Center IVs (HC IVs), 18 Regional Referral Hospitals (RRHs), six (6) Health Center IIIs (HC IIIs), and Health Center IIs (HC IIs), two (2) stand-alone laboratories and One National Referral Hospital (NRH) (See Table 1).

Table 1. Distribution of the sites across the regions, tests done, cases detected, and the number needed to test to detect one case.

| | Number of Health Facilities | | | | | | | | XPERT MTB/RIF | | | | XPERT MTB/RIF ULTRA | | | |
|---|---|---|---|---|---|---|---|---|---|---|---|---|---|---|---|---|
| REGION | HC II | HC III | HC IV | GH | RRH | NRH | Labs | total | TESTS | DS-TB[#] | RR-TB[β] | NNT[*] | TESTS | DS-TB[#] | RR-TB[β] | NNT[*] |
| Kampala | 2 | 3 | 1 | 1 | 5 | 1 | 2 | 15 | 16208 | 1511 | 22 | 11 | 18216 | 1651 | 31 | 11 |
| Mubende | 3 | 0 | 4 | 5 | 1 | 0 | 0 | 13 | 8230 | 748 | 4 | 11 | 13650 | 1077 | 13 | 13 |
| Lira | 0 | 0 | 3 | 1 | 1 | 0 | 0 | 5 | 3290 | 316 | 8 | 10 | 8335 | 826 | 20 | 10 |
| Gulu | 0 | 0 | 3 | 4 | 1 | 0 | 0 | 8 | 5976 | 617 | 11 | 10 | 7929 | 667 | 12 | 12 |
| Arua | 0 | 1 | 1 | 6 | 1 | 0 | 0 | 9 | 9008 | 744 | 19 | 12 | 8604 | 691 | 4 | 12 |
| Moroto | 0 | 0 | 1 | 4 | 1 | 0 | 0 | 6 | 5054 | 751 | 13 | 7 | 5214 | 666 | 11 | 8 |
| Masaka | 0 | 1 | 3 | 1 | 1 | 0 | 0 | 6 | 4755 | 357 | 10 | 13 | 6940 | 434 | 4 | 16 |
| Mbarara | 1 | 1 | 5 | 6 | 2 | 0 | 0 | 15 | 17171 | 1065 | 18 | 16 | 26038 | 1199 | 7 | 22 |
| Fort Portal | 0 | 0 | 2 | 4 | 1 | 0 | 0 | 7 | 5938 | 609 | 5 | 10 | 8049 | 695 | 10 | 12 |
| Hoima | 0 | 0 | 1 | 4 | 1 | 0 | 0 | 6 | 7730 | 640 | 14 | 12 | 8135 | 387 | 11 | 21 |
| Jinja | 0 | 0 | 3 | 4 | 1 | 0 | 0 | 8 | 7243 | 555 | 11 | 13 | 7360 | 702 | 11 | 10 |
| Mbale | 0 | 0 | 2 | 4 | 1 | 0 | 0 | 7 | 4899 | 451 | 8 | 11 | 3622 | 346 | 1 | 10 |
| Soroti | 0 | 0 | 5 | 1 | 1 | 0 | 0 | 7 | 12388 | 443 | 4 | 28 | 6384 | 352 | 9 | 18 |
| TOTAL | 6 | 6 | 34 | 45 | 18 | 1 | 2 | 112 | 107890 | 8807 | 147 | 12 | 128476 | 9693 | 144 | 13 |

[#]DS-TB-Drug Sensitive-Tuberculosis

[β]RR-TB-Rifampicin Resistance-Tuberculosis

[*]NNT-Number Needed to Test

**Table 2. A comparison of tests, Drug Susceptible Tuberculosis (DS-TB) and Multi-Drug Resistant TB (MDR-TB) cases detected by Xpert Ⓡ MTB/RIF and Xpert Ⓡ MTB/RIF Ultra assays.**

|  | Xpert Ⓡ MTB/RIF | Xpert Ⓡ MTB/RIF Ultra | p-value |
|---|---|---|---|
| **Tests** | **107890** | **128476** | **0.73** |
| Mean | 963.34 | 1147.10 |  |
| SD | 680.48 | 842.88 |  |
| **DS-TB[#]** | **8807** | **9693** | **0.30** |
| Mean | 78.63 | 86.54 |  |
| SD | 53.29 | 62.12 |  |
| **RR-TB[*]** | **147** | **144** | **0.95** |
| Mean | 1.31 | 1.28 |  |
| SD | 2.39 | 3.41 |  |
| **NNT[β]** | 12 | 13 |  |
| **Cases Detected/1000** |  |  |  |
| DS-TB | 82 | 75 |  |
| RR-TB | 1 | 1 |  |

[#]DS-TB- Drug susceptible TB

[*]MDR-TB-Multi-Drug Resistant TB

[β]NNT-Number needed to test

### Test performance and TB cases detected

The test performance for Xpert Ⓡ MTB/RIF Ultra assay during the period January to June 2019, 128,476 (M: 1147.10, SD: 842.88) tests were done using Xpert Ⓡ Ultra, detecting 9693 (M: 86.54, SD: 62.12) DS-TB cases and 144 (M: 1.28, SD: 3.41) RR-TB cases. Whilst during the period January to June 2018 when Xpert Ⓡ MTB/RIF was used, 107,890 (M: 963.34, SD: 680.48) tests were done. Of these, 8,807 (M: 78.63, SD: 53.29) were DS-TB cases and 147 (M: 1.31, SD: 2.39) RR-TB cases (Table 1).

### Comparison between Xpert Ⓡ MTB/RIF and Xpert Ⓡ MTB/RIF Ultra

The number needed to test (NNT) to diagnose one DS-TB case was 12 for Xpert MTB/RIF and 13 for Xpert MTB/RIF Ultra, this translated into 75/1000 and 82/1000 DS-TB cases detected with Xpert Ⓡ MTB/RIF Ultra and Xpert Ⓡ MTB/RIF respectively. While both Xpert Ⓡ MTB/RIF Ultra and MTB/RIF did detect 1 MDR-TB case per 1000 tests done. The difference between the two assays in terms of test performance (p = 0.75) and case detection both for susceptible TB (p = 0.31) and MDR-TB (p = 0.95) was not found statistically significant (see Table 2).

### Discussion

We set out to compare the tests done and TB cases detected by Xpert Ⓡ MTB/RIF to Xpert Ⓡ MTB/RIF Ultra assay in Uganda. Xpert Ⓡ MTB/RIF Ultra was found neither to produce significantly more tests nor more tuberculosis cases either DS-TB or RR-TB when compared to Xpert Ⓡ MTB/RIF assay.

The switch from Xpert Ⓡ MTB/RIF to Xpert Ⓡ MTB/RIF Ultra did increase the percentage of tests done using the same platform by up to 19%. However, given the significant reduction in the turn-around time of the Xpert Ⓡ MTB/RIF Ultra compared to the Xpert Ⓡ MTB/RIF [9], we expected an increment of up 50% [10]. We do recommend that the health systems approach is used to elucidate all the probable potential of the MTB/RIF Ultra assay.

There were 7 TB cases per 1000 tests done reduction on changing the assay from Xpert®ᵣ MTB/RIF to Xpert®ᵣ MTB/RIF Ultra. A modeling mathematical study had found the reverse, and it anticipated 2 to 9 TB cases per 1000 individuals tested when the switch was made from MTB/RIF to Xpert®ᵣ MTB/RIF Ultra [12]. Therefore, we recommend further studies to evaluate the incremental benefit of switching from Xpert®ᵣ MTB/RIF Ultra to Xpert®ᵣ MTB/RIF in detecting additional TB cases.

Xpert®ᵣ MTB/RIF Ultra did detect the same number of MDR-TB cases as Xpert®ᵣ MTB/RIF. This agrees with most of the systematic reviews comparing Xpert®ᵣ MTB/RIF and Xpert®ᵣ MTB/RIF Ultra. However, given that over 70% of the estimated RR-TB cases are still being missed globally [13], there is a need for a faster point of care, with a high sensitivity to diagnose RR-TB accurately thereby increasing the number of people diagnosed with RR-TB, hence reducing its associated morbidity and mortality.

This study was not without limitations, some data which could have been beneficial in the comparison (Number with DS-TB and RR-TB who are either children or HIV positive) since Xpert®ᵣ MTB/RIF was found to have a significant difference in these sub-groups were not available. It also could have been beneficial to compare smear positive and smear-negative samples, since Xpert®ᵣ MTB/RIF Ultra is more sensitive in the latter, more TB cases would have been detected. We also did not control for the historical threats to validity which could have been caused by a change in management and personnel. However, this was minimized by reducing the time between the before and after periods,

## Conclusion and recommendations

The study found no significant difference in test performance and overall MTB case detection. The health systems approach should be used to elucidate all the probable potential of the Xpert®ᵣ MTB/RIF Ultra assay transitioning from the Xpert®ᵣ MTB/RIF assay.

## Supporting information

**S1 Dataset.**
(XLSX)

## Acknowledgments

We are grateful to the Uganda National TB and Leprosy Programme which provided us with the data and permitted us to use the data. As well as the National Tuberculosis Reference Laboratory who are the main custodians of the Tuberculosis Laboratory data.

## Author Contributions

**Conceptualization:** Michael Kakinda, Didas Tugumisirize, Abdunoor Nyombi.

**Data curation:** Michael Kakinda, Marvin Mugisha, Stavia Turyahabwe, Joseph K. B. Matovu.

**Formal analysis:** Michael Kakinda, Marvin Mugisha, Stavia Turyahabwe, Simon Walusimbi.

**Investigation:** Michael Kakinda, Didas Tugumisirize.

**Methodology:** Michael Kakinda, Didas Tugumisirize, Abdunoor Nyombi, Stavia Turyahabwe, Simon Walusimbi, Joseph K. B. Matovu.

**Software:** Michael Kakinda.

**Supervision:** Michael Kakinda, Abdunoor Nyombi, Stavia Turyahabwe, Simon Walusimbi, Joseph K. B. Matovu.

**Validation:** Michael Kakinda, Didas Tugumisirize, Abdunoor Nyombi, Stavia Turyahabwe, Simon Walusimbi, Joseph K. B. Matovu.

**Visualization:** Michael Kakinda, Didas Tugumisirize, Joseph K. B. Matovu.

**Writing – original draft:** Michael Kakinda, Didas Tugumisirize, Simon Walusimbi, Joseph K. B. Matovu.

**Writing – review & editing:** Michael Kakinda, Didas Tugumisirize, Abdunoor Nyombi, Marvin Mugisha, Stavia Turyahabwe, Simon Walusimbi, Joseph K. B. Matovu.

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
