## [Decision Letter · Decision Letter 0]

14 Jan 2021

PONE-D-20-34522

Comparison of tests done and Tuberculosis cases detected by Xpert® MTB/RIF and Xpert® MTB/RIF-Ultra in Uganda

PLOS ONE

Dear Dr. Kakinda,

Thank you for submitting your manuscript to PLOS ONE. After careful consideration, we feel that it has merit but does not fully meet PLOS ONE’s publication criteria as it currently stands. Therefore, we invite you to submit a revised version of the manuscript that addresses the points raised during the review process.

We look forward to receiving your revised manuscript.

Kind regards,

Shampa Anupurba, MD

Academic Editor

PLOS ONE

Additional Editor Comments:

In this manuscript Xpert MTB/RIF has been compared with Xpert MTB/RIF Ultra by including the data from 112 sites during two different time periods.Although it has been concluded that there was no significant difference in test performance and overall detection of DS-TB and RR-TB, the study could have highlighted the difference between smear positive and negative samples. Secondly,it would have been interesting to note whether there were more cases from any particular extra pulmonary sample.There have been studies which have shown that sensitivity and specificity varied according to the type of sample.Thirdly, the performance of Xpert MTB/RIF ultra in paediatric age group could have been shown.

Minor corrections

Line 75- The sentence could be completed after 'Jan2012'

Line 110- Rifampicin resistant TB should be RR-TB

4th and 5th lines under 'Comparison between Xpert® MTB/RIF and Xpert® MTB/RIF Ultra'- MDR-TB to be replaced by RR-TB. Also, in Table 2, MDR-TB to be replaced by RR-TB.

3. You indicated that ethical approval was not necessary for your study. We understand that the framework for ethical oversight requirements for studies of this type may differ depending on the setting and we would appreciate some further clarification regarding your research. Could you please provide further details on why your study is exempt from the need for approval and confirmation from your institutional review board or research ethics committee (e.g., in the form of a letter or email correspondence) that ethics review was not necessary for this study? Please include a copy of the correspondence as an "Other" file.

"No."

Reviewers' comments:

Reviewer's Responses to Questions

**Comments to the Author**

1. Is the manuscript technically sound, and do the data support the conclusions?

Reviewer #1: Yes

2. Has the statistical analysis been performed appropriately and rigorously? 

Reviewer #1: Yes

3. Have the authors made all data underlying the findings in their manuscript fully available?

Reviewer #1: Yes

4. Is the manuscript presented in an intelligible fashion and written in standard English?

Reviewer #1: Yes

5. Review Comments to the Author

Reviewer #1: This is a simple and well-written study. However, information about the type of sample, type of TB (pulmonary, extrapulmonary with disease site), children vs adults should be included for a better comparison.

Furthermore, information should be provided about the issue of “trace positive samples” by Ultra. In addition to detection of the single-copy rpoB gene for the simultaneous diagnosis of TB and rifampicin resistance, Ultra includes two new and more sensitive PCR assays that target the multicopy IS6110 and IS1081 genes to improve M.tuberculosis detection in paucibacillary samples. Information about RIF-R is not available in IS6110 or IS1081 positive and rpoB negative samples are categorised as “trace” by Ultra. Thus MDR-TB cases can be missed among the “trace” positive samples. For these samples, information about drug resistance can only be provided by culture. Therefore, when comparing the two methods, it is important to have this information.

Minor comments:

Mycobacterium Tuberculosis should be written as Mycobacterium tuberculosis

The sentence in the last paragraph of discussion should be corrected “This study was not with limitations….”

6. PLOS authors have the option to publish the peer review history of their article (what does this mean?). If published, this will include your full peer review and any attached files.

Reviewer #1: No

---

## [Author Response · Author response to Decision Letter 0]

16 Jul 2022

POINT-BY-POINT RESPONSE TO THE EDITOR’S AND THE REVIEWERS’ COMMENTS

Response to the Editor’s comments

In this manuscript Xpert MTB/RIF has been compared with Xpert MTB/RIF Ultra by including the data from 112 sites during two different periods. Although it has been concluded that there was no significant difference in test performance and overall detection of DS-TB and RR-TB, the study could have highlighted the difference between smear positive and negative samples. Secondly, it would have been interesting to note whether there were more cases from any extrapulmonary sample. There have been studies that have shown that sensitivity and specificity varied according to the type of sample. Thirdly, the performance of Xpert MTB/RIF ultra in the pediatric age group could have been shown.

We thank the editor for this comment since it is fundamental to why the Xpert MTB/RIF was introduced. The authors have tried to accommodate the comment in the limitations section since it is not routine to have a culture for all Sputum TB samples. Unless they are Rifampicin Resistant Indeterminate. The same applies to the extra-pulmonary versus the pulmonary samples, the former samples are likely to be negligible. So, there are unlikely to change the overall results. 

Line 75- The sentence could be completed after 'Jan2012'

We have made this correction. 

Line 110- Rifampicin resistant TB should be RR-TB

This correction has been made. 

4th and 5th lines under 'Comparison between Xpert® MTB/RIF and Xpert® MTB/RIF Ultra'- MDR-TB to be replaced by RR-TB. Also, in Table 2, MDR-TB is to be replaced by RR-TB.

This correction has been made, see lines 4 and 5 and Table 2. 

Response: We have revised our manuscript to ensure that the level 1 headings are 18 pt font, while level two headings are 16 pt font and level 3 are 14 pt font. They are all in bold and we have capitalized only the first word of the heading, the first word of the sub-heading and any proper nouns and genus names. The rest of the text is 12 pt font. Supplementary information files have been renamed following the PLOS ONE’s style requirements. 

The authors have edited the manuscript using commercially available software editors, and all the authors did edit the manuscript. We have provided a manuscript with tracked changes and a clean copy of the edited manuscript. 

3. You indicated that ethical approval was not necessary for your study. We understand that the framework for ethical oversight requirements for studies of this type may differ depending on the setting and we would appreciate some further clarification regarding your research. Could you please provide further details on why your study is exempt from the need for approval and confirmation from your institutional review board or research ethics committee (e.g., in the form of a letter or email correspondence) that ethics review was not necessary for this study? Please include a copy of the correspondence as an "Other" file.

We have made changes for the ethical approval section. I have attached a letter from the Operational Research and Ethics Review Committee of the National Tuberculosis and Leprosy Program of Uganda. The authors had previously got the approval to use the data and publish it but, it had not been written. However, we got a copy and will attach it under others. 

Response: The following statement has been used to meet the ‘Data availability’ requirements of the journal:

“All relevant data are within the paper and its Supporting Information files”

"No"

a. Please clarify the sources of funding (financial or material support) for your study. List the grants or organizations that supported your study, including funding received from your institution.

b. State what role the funders took in the study. If the funders had no role in your study, please state: “The funders had no role in study design, data collection, and analysis, decision to publish, or preparation of the manuscript.”

d. If you did not receive any funding for this study, please state: “The authors received no specific funding for this work.”

Response: The following financial disclosure information statement should be included in the online submission form: 

“There was no grant or funds provided for this study”…

6. Thank you for stating the following in your Competing Interests section: 

"No"

Please complete your Competing Interests on the online submission form to state any Competing Interests. If you have no competing interests, please state "The authors have declared that no competing interests exist.", as detailed online in our guide for authors at http://journals.plos.org/plosone/s/submit-now. This information should be included in your cover letter; we will change the online submission form on your behalf.

Response: The following statement should be included in the online submission system:

“The authors have declared that no competing interests exist”.

7. In your Data Availability statement, you have not specified where the minimal data set underlying the results described in your manuscript can be found. PLOS defines a study's minimal data set as the underlying data used to reach the conclusions drawn in the manuscript and any additional data required to replicate the reported study findings in their entirety. All PLOS journals require that the minimal data set be made fully available. For more information about our data policy, please see http://journals.plos.org/plosone/s/data-availability.

Response: The following statement has been used to meet the ‘Data availability’ requirements of the journal:

“All relevant data are within the paper and its Supporting Information files”

Response to Reviewer’s comments

Comment: Furthermore, information should be provided about the issue of “trace positive samples” by Ultra. In addition to detection of the single-copy rpoB gene for the simultaneous diagnosis of TB and rifampicin resistance, Ultra includes two new and more sensitive PCR assays that target the multicopy IS6110 and IS1081 genes to improve M.tuberculosis detection in paucibacillary samples. Information about RIF-R is not available in IS6110 or IS1081 positive and rpoB negative samples are categorized as “trace” by Ultra. Thus MDR-TB cases can be missed among the “trace” positive samples. For these samples, information about drug resistance can only be provided by culture. Therefore, when comparing the two methods, it is important to have this information.

We have acknowledged the shortcomings of not having the data for culture and we have included it in the study limitations. However, in the literature despite the changes in the targets on IS6110 and IS1081 genes, there wasn’t a significant change in the detection of Rifampicin Resistant Cases. 

Comment: Mycobacterium Tuberculosis should be written as Mycobacterium tuberculosis

We thank the reviewer for this comment, and this has been revised.

Comment:The sentence in the last paragraph of the discussion should be corrected “This study was not with limitations….”

We thank the reviewer for this comment. The authors have corrected this to “…This study was not without limitations”…

---

## [Editor Report · Decision Letter 1]

12 Sep 2022

PONE-D-20-34522R1Comparison of tests done and Tuberculosis cases detected by Xpert® MTB/RIF and Xpert® MTB/RIF-Ultra in UgandaPLOS ONE

Dear Dr. Kakinda,

Thank you for submitting your manuscript to PLOS ONE. After careful consideration, we feel that it has merit but does not fully meet PLOS ONE’s publication criteria as it currently stands. Therefore, we invite you to submit a revised version of the manuscript that addresses the points raised during the review process.

The manuscript has been revised satisfactorily. However, there are two minor corrections

Line 58- 'Tuberculosis' to be replaced by 'tuberculosis'

Line 137- 'got' to be replaced by 'obtained'

We look forward to receiving your revised manuscript.

Kind regards,

Shampa Anupurba, MD

Academic Editor

PLOS ONE
---

## [Author Response · Author response to Decision Letter 1]

15 Sep 2022

POINT-BY-POINT RESPONSE TO THE EDITOR’S AND THE REVIEWERS’ COMMENTS

Response to the Editor’s Comments 

The manuscript has been revised satisfactorily. However, there are two minor corrections

Line 58- 'Tuberculosis' to be replaced by 'tuberculosis'

We do thank the editor for this observation. We have made this correction. 

Line 137- 'got' to be replaced by 'obtained

We have made this correction.

---

## [Editor Report · Decision Letter 2]

27 Sep 2022

Comparison of tests done and Tuberculosis cases detected by Xpert® MTB/RIF and Xpert® MTB/RIF-Ultra in Uganda

PONE-D-20-34522R2

Dear Dr. Kakinda,

We’re pleased to inform you that your manuscript has been judged scientifically suitable for publication and will be formally accepted for publication once it meets all outstanding technical requirements.

Kind regards,

Shampa Anupurba, MD

Academic Editor

PLOS ONE
---

## [Editor Report · Acceptance letter]

29 Sep 2022

PONE-D-20-34522R2 

Comparison of tests done, and Tuberculosis cases detected by Xpert® MTB/RIF and Xpert® MTB/RIF-Ultra in Uganda 

Dear Dr. Kakinda:

I'm pleased to inform you that your manuscript has been deemed suitable for publication in PLOS ONE. Congratulations! Your manuscript is now with our production department. 

Kind regards, 

on behalf of

Dr. Shampa Anupurba 

Academic Editor

PLOS ONE